# Modeling and Cutting Mechanics in the Milling of Polymer Matrix Composites

**DOI:** 10.3390/ma18133017

**Published:** 2025-06-25

**Authors:** Krzysztof Ciecieląg, Andrzej Kawalec, Michał Gdula, Piotr Żurek

**Affiliations:** 1Department of Production Engineering, Faculty of Mechanical Engineering, Lublin University of Technology, Nadbystrzycka 36, 20-618 Lublin, Poland; 2Department of Manufacturing Techniques and Automation, Faculty of Mechanical Engineering and Aeronautics, Rzeszów University of Technology, al. Powstańców Warszawy 12, 35-959 Rzeszów, Poland; ak@prz.edu.pl (A.K.); gdulam@prz.edu.pl (M.G.); p_zurek@prz.edu.pl (P.Ż.)

**Keywords:** milling, cutting forces, glass fiber-reinforced plastics, carbon fiber-reinforced plastics, tool geometry

## Abstract

The study investigates the problem of modeling cutting-force components through response surface methodology and reports the results of an investigation into the impact of machining parameters on the cutting mechanics of polymer–matrix composites. The novelty of this study is the modeling of cutting forces and the determination of mathematical models of these forces. The models describe the values of forces as a function of the milling parameters. In addition, the cutting resistance of the composites was determined. The influence of the material and rake angle of individual tools on the cutting force components was also determined. Measurements of the main (tangential) cutting force showed that, using large rake angles for uncoated carbide tools, one could obtain maximum force values that were similar to those obtained with polycrystalline diamond tools with a small rake angle. The results of the analysis of the tangential component of cutting resistance showed that, regardless of the rake angle, the values range from 140 N to 180 N. An analysis of the feed component of cutting resistance showed that the maximum values of this force ranged from 46 N to 133 N. The results showed that the highest values of the feed component of cutting resistance occurred during the machining of polymer composites with carbon fibers and that they were most affected by feed per tooth. It was also shown that the force models determined during milling with diamond insert tools had the highest coefficient of determination in the range of 0.90–0.99. The cutting resistance analysis showed that the values tested are in the range of 3.8 N/mm^2^ to 15.5 N/mm^2^.

## 1. Introduction

Polymer composites are employed wherever components must combine low specific mass with high mechanical strength [1]. These materials also exhibit high corrosion resistance, vibration-damping capability, and good electrical-insulation properties. Structurally, they consist of at least two constituents that form a heterogeneous, anisotropic system exhibiting direction-dependent physical properties [2]. Due to these advantageous characteristics, composite materials are widely adopted in applications where product performance, rather than cost, dictates material selection. Glass and carbon fiber-reinforced composites, saturated with epoxy resin, are used predominantly in the aerospace, space, and automotive sectors. Their resistance to aggressive environments further enhances their suitability for use in medical devices. Composite materials are also used for the regeneration and protection of surfaces against the effects of chemical substances occurring in the form of solids, liquids or gases. Often, the repaired surface of the composite is characterized by better parameters than that made of steel.

The variety of applications and advantageous properties of composites require the machining process to achieve the required shape and dimensional tolerances [3]. The primary methods of machining for polymer composites include turning, drilling and milling [4]. Machining operations performed on polymer composites are used to make cut-outs of complex shapes, make recesses and grooves, as well as to make and deepen holes. In milling operations, the side and front surfaces are machined to obtain appropriate roughness. The implementation of the milling process for polymer composites is related to the selection of machining conditions [5].

One of the factors influencing the milling process is the selection of technological parameters, which depends on the material structure, fiber orientation, fiber content in the composite, fiber type, and desired surface roughness [6]. The typical range of cutting speeds used for machining polymer composites reinforced with glass fibers (GFRPs) and carbon fibers (CFRPs) ranges between 50 and 250 m/min [7,8], although this range is often extended up to 500 m/min [9]. As for feed per tooth, the recommended range is from 0.05 to 0.5 mm/tooth [7,8,10]. The third milling parameter, the depth of cut, is recommended to be within 0.1 and 4 mm [9,11,12]. Studies of the influence of milling parameters on polymer composites have shown that feed per tooth has the greatest impact on the surface quality after machining, with cutting speed being the second most significant parameter [11]. Studies have also demonstrated that the depth of cut does not significantly affect the machining quality of polymer composites. An increase in cutting speed, particularly in the case of glass fiber-reinforced plastics, results in a rise in thermal energy, which is primarily absorbed by the tool, thereby accelerating tool wear [13].

Another vital aspect of milling polymer composites is the selection of the tool [14]. The choice of tools depends on the type of material being machined, the expected surface quality after machining and the size of a production batch. Typically, diamond plate tools are employed for milling polymer composites in larger production batches [2]. These tools are highly resistant to abrasion and maintain a sharp cutting edge for an extended period. Tools made of coated cemented carbides are also used. TiAlN-coated cemented carbides reduce the coefficient of friction during milling, which, combined with increased abrasion resistance of the tool, results in a longer service life of the blade [2]. Uncoated carbide tools wear out quickly suitable for small production batches where machining costs are a critical factor [15]. Ceramic tools are not used for machining polymer composites due to their heterogeneous nature. Ceramic tools have low strength and are brittle, which makes them highly susceptible to impact damage, and their very low thermal conductivity prevents effective dissipation of heat generated during machining [3]. Tools for machining polymer composites should be very sharp to ensure rapid material separation by cutting. Additionally, tool geometry is crucial due to its influence on cutting forces and surface defect formation [15].

Results of previous studies on machining polymer composites and the effects of technological parameters of this process differ [16]. This is primarily related to the heterogeneous structure of these materials, the occurrence of delamination, surface damage, variable roughness, and fiber pull-out [12]. Defects such as delamination that arise during milling can lead to the disqualification of the material from further use [17]. The study examined the delamination of the upper layers of composite material, which leads to the rejection of components. The results of the study allow for the adjustment of the processing methodology aimed at reducing the number of such defects. Many studies on machining polymer composites focus on selecting the most appropriate milling parameters, proper fiber orientation, and choosing tools with suitable geometry and costs adequate to the expected results [10,18]. However, the overriding factor is to minimize problems and defects that occur during milling [19,20].

The selection of cutting parameters using the Taguchi method and analysis of variance (ANOVA) showed that the feed rate had the most significant effect on cutting forces and surface roughness [21]. The impact of increasing feed rate on the deterioration of surface roughness was also confirmed in another study [22]. It was found that low cutting speed, i.e., low tool revolutions, contributes to surface damage [22]. This statement was also confirmed during the study of face milling, where it was demonstrated that low feed rates at high rotational speeds were conducive to low surface roughness [23]. However, a study conducted with an artificial neural network showed that the use of low feed values and low rotational speed was effective mainly when machining glass fiber-reinforced plastics [23]. The application of neural networks and ANOVA in another study focused on the milling process revealed that cutting speed and depth of cut were the most significant factors influencing delamination and surface roughness of glass fiber-reinforced polymer composites [24]. It was demonstrated that the depth of cut had the primary impact on surface roughness. In contrast, the cutting speed had the most significant effect on delamination during the machining of glass fiber-reinforced plastics [24]. Besides examining the influence of machining parameters, a relationship between surface quality and tools was also established. It was found that the use of a small helix angle, medium cutting speeds and low feed rates contributed to reducing surface roughness [25].

In the milling process, cutting forces are a significant physical quantity [26]. They are influenced by various factors, including machining parameters and the interaction of physical phenomena at the contact zone between the workpiece and the tool [27]. Due to the ease of measuring cutting forces, many studies established relationships between cutting forces and machining parameters by selecting appropriate conditions [28]. It was found that increasing the feed rate increases the cutting forces, while the cutting speed reduces their values. Unlike in the machining of widely used engineering materials such as steel or aluminum alloys, these forces depend on the fiber orientation. Due to the abrasive nature of fibers, diamond tools can chip, resulting in accelerated wear of the blade. As a consequence, cutting forces and surface roughness increase. In the case of composites, the type of fiber used is an important factor influencing cutting forces. Carbon fibers, compared to glass fibers, offer greater cutting resistance, resulting in higher forces. In terms of geometry, the corner rounding radius is important. Large corner rounding radii result in higher cutting forces due to the larger cross-sectional area of the cut layer. The diamond coating also increases tool radius, which leads to an increase in milling forces [10]. The effect of fiber hardness is a high cutting force, which is one of the causes of tool wear in the machining of composite materials [2]. The machining of composite materials using carbide tools causes lateral surface wear, which increases significantly at high cutting speeds. Milling studies showed that during the machining of polymer composites reinforced with carbon fibers, the cutting force increases nonlinearly with the feed rate. It was also found that the change in cutting speed did not correlate with the change in cutting force [29]. Studies on glass fiber-reinforced polymer composites investigating the influence of tool helix angle, fiber orientation angle and face milling parameters on the cutting force showed that the cutting force increased with higher feed rates and fiber orientation angles but decreased with increasing tool helix angle [30]. A comparison of the cutting forces during machining of glass fiber-reinforced plastics (GFRPs) and carbon fiber-reinforced plastics (CFRPs) under the same machining parameters showed higher forces in CFRP machining [31]. In terms of tool effect, it was revealed that the use of a rake angle close to 0° and a high cutting speed, as well as a rake angle of 5° and a low cutting speed, produced minimal surface roughness. The tool geometry also affected on the cutting force, which was the weakest for the tool rake angles in the range of 0–10° [32].

The use of milling process models is quite common when machining homogeneous materials such as steels, aluminum alloys [33] and titanium alloys or nickel alloys, which are generally challenging to machine [34,35,36], especially if the change in material properties of these alloys at elevated temperatures is taken into account [37]. Problems arise when the workpiece is made of a polymeric composite with a heterogeneous structure [38]. In this case, the model is adapted to the composite material by considering cutting forces that describe the resistance of composite fibers. In addition, oscillations are introduced between the tool and the workpiece to reduce vibrations [39]. The behavior of material during machining is an input factor for preparing models for numerical simulations. This is because research on the machining process for composite materials is expensive. The cost of polymeric composites for experiments is considerable [40]. Hence, approaches based on micromechanics, equivalent homogeneous material, or a combination of these two concepts, are used [41]. On the one hand, the approach based on micromechanics enables the study of physical phenomena in a local manner, but on the other, it increases CPU time [42]. In turn, the approach based on the equivalent homogeneous material reduces the CPU time, but its disadvantage is that it is unable to predict the damage observed at the fiber–matrix interface [43]. Therefore, to improve the quality of simulation, simulation capabilities are being expanded by combining the approaches based on micromechanics and the equivalent homogeneous material. In effect, it is possible to simulate cutting forces and defects in the product [44]. A further development of these techniques enabled the simulation and investigation of the influence of fiber orientation and chip formation processes [45]. Initially, simulations on machining polymer composites did not always match the experimental observations [46]. Only after introducing numerical factors based on micromechanical models, which took into account matrix and fiber properties, was it possible to predict cutting forces [42]. However, this approach limited force predictions to only parts of the fibers. For this reason, a simulation was performed in which composite fibers were treated as elastic elements [45]. The disadvantage of this approach is, however, the lack of consideration of imperfections in the strength of the fibers. Another approach to numerical analysis involved the use of both micro- and macro-approaches, which allowed the results to be closer to the experimental model [40]. The development of computer techniques requires knowledge of the cutting mechanics of composite materials at a micro-level.

Given the differing and often inconsistent results of previous studies on milling polymer composites, it is essential to investigate the cutting process using the response surface methodology (RSM) in a central composite rotatable design. The aim of this work is therefore to determine parameters that have the most positive impact on cutting forces in the milling process conducted with variable technological parameters. Two different composite materials are analyzed in the study, and they are milled using three cutting tools of different geometries and materials.

## 2. Materials and Methods

### 2.1. Materials and Sample Preparation

Two types of samples made from polymer composites were used in the study. One sample was a glass fiber-reinforced polymer composite named EGL/EP 3200-120, impregnated with epoxy resin. The other was a carbon fiber-reinforced polymer composite named HexPly AG193PW/3501/6SRC41, also impregnated with epoxy resin. The samples were in the form of plates with the dimensions of 10 × 50 × 150 mm. The composite materials consisted of 40 layers of prepregs (each 0.25 mm thick) arranged in a 0°/90° lay-up configuration. This configuration provides strength in two perpendicular axes of the material. The materials were prepared using the autoclave technique. Before putting them in the autoclave for 2-h curing at 177 °C (±2 °C) with a pressure of 0.3 MPa, the samples were conditioned in a special room. The room had a temperature ranging from 18 °C to 30 °C and humidity below 60%. Additionally, the air cleanliness in this environment was controlled to ensure that the number of solid particles did not exceed 10,000 per cubic meter. Figure 1 shows the samples and their dimensions, together with the mesh density of the composite materials. Glass and carbon fibers reinforce polymer composites. They are arranged in a 0–90° pattern to form a fabric which, when saturated with epoxy resin, forms a prepreg. This arrangement ensures balanced material strength, which is particularly important in machining.

### 2.2. Tools

The milling process was performed using a 12 mm nominal diameter indexable cutter, consisting of a body with the symbol R217.69-1212.0-06-2AN, in which cutting inserts were mounted. Three different types of cutting inserts from SECO were used: XOEX060204FR-E03 H15 (uncoated carbide insert), XOMX060204R-M05 F40M (PVD TiAlN-coated carbide insert, where PVD coating stands for physical vapor deposition coating) and XOEX060204FR PCD05 (polycrystalline diamond insert). Each tool had a different rake angle γ ranging from 8° to 29.2°. Table 1 provides the specifications of the tools used in the tests.

Figure 2a–c show the rake and clearance angles for the three tested cutting tools.

### 2.3. Method and Technological Parameters of the Milling Process

Milling operations of polymer composites were carried out on a DMG DMU 100 monoBLOCK milling center (Rzeszów, Poland). A hydraulic machine vice was mounted in the milling center workspace. The composite samples were fixed in the jaws of the vice. Cutting forces were measured using a Kister 9123C (Rzeszów, Poland) piezoelectric rotating four-component dynamometer with a maximum rotational speed of 10,000 rev/min. The piezoelectric rotating dynamometer was mounted directly in the spindle of the milling center. The dynamometer was connected to a four-channel charge amplifier, Kistler 5223B. Combined with a KUSB-3100 data acquisition board as a measurement card and quickDAQ software, it enabled the recording of cutting force curves. The tests were conducted with a sampling frequency of 10 kHz. The experimental setup is shown in Figure 3.

To eliminate noise, the obtained cutting force signals were filtered using a Chebyshev low-pass filter, The filter was implemented using a proprietary algorithm and program developed in MATLAB (vR2022b).

A coordinate system of the cutting force components is shown in Figure 4. Directions of the cutting force components were defined in a tool coordinate system that was consistent with the actual machining setup. The force F_x_ is the feed component of cutting resistance. The other component, F_y_, is the main cutting force, i.e., the tangential component of cutting resistance acting in the direction of the main motion.

Figure 4 also shows the values of a_p_, i.e., the depth of cut, and a_e_, i.e., the width of cut that was equal to the diameter of the milling cutter. Complete symmetric milling was performed to prevent the effect of overlapping machined bands on force signal characteristics. Therefore, the study focuses on forces solely generated by the engagement of the cutter edges with the workpiece material. This is, the forces resulting exclusively from the kinematics of the milling process.

Technological parameters of milling were determined using the response surface methodology (RSM) in a central composite rotatable design, as implemented in Statistica 13.3 software.

The statistical method of determining the response surface (RSM) was used to model the relationships between the cutting process parameters and the cutting force components. RSM-based research plans refer to the determination of a response surface based on a general Equation (1) of the type: Y = β_0_ + β_1_X_1_ + β_2_X_2_ + β_11_(X_1_)^2^ + β_22_(X_2_)^2^ + β_12_X_1_X_2_,(1)

As a result, a model is fitted to the experimental output values that captures the influence of the main input values (X_1_, X_2_), the interactions between the input values (X_1_X_2_), and the quadratic terms (X_1_^2^, X_2_^2^). β_0_ is the intercept of the arithmetic averages of all the quantitative outcomes. Parameters β_1_, β_2_, β_11_, β_22_ and β_12_ are the coefficients computed from the experimental values of Y. Regression statistics commonly used in mechanical engineering, e.g., ANOVA, were used to identify the most influential factors, create suitable mathematical model and evaluate its fit to the measurement data.

The experiments were conducted with a constant depth of cut set at 1 mm. Variable cutting speed and feed per tooth were used in the tests. Values of these parameters are listed in Table 2.

The aim of the experiments was to determine the response surface using a rotatable design with star points (shown in Figure 5).

The applied range of cutting speeds used allowed us to determine the central value at 210 m/min. The iteration was set at 110 m/min, which allowed us to obtain point values at 100 m/min and 320 m/min. The length of the square diagonal was 311 m/min (d = a√2, where a = 320 m/min–100 m/min), from which the star point values were 54.5 m/min and 365.5 m/min (values that were half lower and higher the diagonal than the central point value). As for the second parameter, i.e., feed per tooth, the central value was set at 0.3 mm/tooth. After determining the iteration, points with values of 0.15 mm/tooth and 0.5 mm/tooth were obtained. The length of the square diagonal was 0.49 mm/tooth and the star points were 0.05 mm/tooth and 0.54 mm/tooth.

## 3. Results

An analysis of cutting forces during milling of polymer–matrix composites provides insight into the mechanics of material removal, accounting for the combined properties of a workpiece–tool material pair and the cutting tool geometry. In the present study, two types of polymeric composites were machined using three milling cutters that differed in both tool material and rake angle. Experiments were conducted using variable technological parameters, specifically feed per tooth and cutting speed. The assessment of the machining parameters was based on modelling and evaluation of individual components of the cutting force. Representative force–time histories for the F_x_ and F_y_ components recorded during GFRP machining with polycrystalline-diamond inserts are shown in Figure 6a,b.

The obtained cutting-force data made it possible to construct response surfaces for the F_x_ and F_y_ components in a tool coordinate system. Since polymer–matrix composites are inherently heterogeneous, their cutting force levels differ considerably from those observed in machining steels or aluminum alloys.

Figure 7a,b shows the response surfaces obtained in milling glass fiber-reinforced plastics with uncoated cemented-carbide cutters. The plots illustrate the effect of feed per tooth and cutting speed on the feed force F_x_ and the main (tangential) force F_y_.

An analysis of the surface in Figure 7a reveals that the highest F_x_ force of 58 N was obtained with the highest feed per tooth and cutting speed. The surface in Figure 7b shows that the highest values of the F_y_ component, ranging from 152 N to 158 N, were achieved with the feed value of 0.54 mm/tooth in the entire analyzed range of cutting speed. A comparison of the two cutting force components shows higher values of the main (tangential) force F_y_. The high values of F_y_ obtained with the highest tested feed per tooth indicate an increase in the cross-sectional area of the removed layer. In addition, the force fluctuations visible in Figure 6a,b suggest a temporary accumulation of the material in front of the cutting edge as a layer of the heterogeneous composite is removed. Reducing the feed per tooth mitigates the principal force, thereby lessening the mechanical load on the uncoated cemented-carbide tool.

An ANOVA analysis was performed to select appropriate models and establish formulas describing the cutting force components during GFRP machining with uncoated carbide tools:F_x_ = 23.69 + 0.067v_c_ − 0.00027 v_c_^2^ − 6.036f_z_ − 64.73f_z_^2^ + 0.33v_c_f_z_, (2)F_y_ = 10.14 + 0.075v_c_ − 0.00013 v_c_^2^ + 315.53f_z_ − 122.96f_z_^2^ − 0.0013v_c_f_z_,(3)

Figure 8a,b shows the response surfaces obtained from the relations in (4) and (5), describing the effects of cutting speed and feed on the cutting force components F_x_ and F_y_, respectively.

The highest value of the feed force F_x_ occurs when the highest values of feed per tooth and cutting speed are used (Figure 8a), this force value amounting to 131 N, which is almost twice the value obtained with the uncoated carbide tool. However, the uncoated carbide tool had a rake angle of 29.2° and the coated carbide tool had an angle of 20.2°. An analysis of the tangential component F_y_ shows that the highest forces are obtained for the lowest and highest cutting speeds and at the highest feed per tooth (Figure 8b). A decrease in the value of the force F_y_ is visible in the middle range of the applied cutting speeds. The range of the force F_y_ values for the TiAlN-coated carbide tools is similar to the values obtained for the uncoated carbide cutting tools. 

Formulas for calculating the cutting forces in GFRP machining with TiAlN-coated carbide tools:F_x_ = 14.20 + 0.28v_c_ − 0.00043v_c_^2^ + 14.16f_z_ − 85.14f_z_^2^ + 0.42v_c_f_z_,(4)F_y_ = 22.60 − 0.11v_c_ + 0.00087v_c_^2^ + 394.70f_z_ − 143.23f_z_^2^ − 0.48v_c_f_z_, (5)

The next stage of the cutting-force analysis for the glass fiber-reinforced polymer composite involved the use of a tool fitted with polycrystalline diamond inserts with an 8° rake angle. Response surfaces constructed from Equations (6) and (7) and plotted in Figure 9a,b reveal that this tool generates the lowest feed-force component F_x_ of all the cutters, with a minimum of about 46 N (Figure 9a). The surface in Figure 9b clearly shows that the highest tangential forces F_y_ occur at the maximum feed per tooth; under this condition, F_y_ ranges roughly from 160 N to 184 N.

Formulas for the cutting forces in GFRP machining with polycrystalline diamond insert tools:F_x_ = −21.95 + 0.30v_c_ − 0.00055v_c_^2^ + 116.35f_z_ − 129.69f_z_^2^ + 0.02v_c_f_z_,(6)F_y_ = −1.95 + 0.19v_c_ − 0.00034v_c_^2^ + 395.88f_z_ − 205.29f_z_^2^ − 0.05v_c_f_z_, (7)

To sum up, an analysis of the cutting forces for the glass fiber-reinforced composite material demonstrates that the rake angle and tool material are of vital importance when machining non-homogeneous materials. Tools with polycrystalline diamond inserts with a rake angle of 8° yielded the lowest force components in the feed motion direction. Still, these values were also similar to those obtained during GFRP machining using uncoated cemented carbides (with a rake angle over three times larger, i.e., 29.2°). The results enable us to conclude that it is possible to use less expensive tools with uncoated cemented carbides if we aim to achieve low feed forces at low tool costs. An important conclusion regarding the main (tangential) force in GFRP milling using tools with different rake angles is that the obtained force values are similar. This phenomenon indicates that by reducing tool costs (the cost of a polycrystalline diamond tool is almost eight times higher than that of uncoated carbide tools) while using large rake angles, similar values of the main (tangential) force component can be obtained. The cost of a single cutting insert made of polycrystalline diamond is approximately €100, while the cost of a carbide insert is approximately €12.

An analysis of the cutting force components was also carried out for the polymer composite with carbon fibers. Examples of the cutting force components F_x_ and F_y_ are plotted in Figure 10a,b.

The next stage of the analysis was to determine the response surface for CFRP. Figure 11a,b show the effect of feed per tooth and cutting speed on the F_x_ and F_y_ components of the cutting force for CFRP. The F_x_ force results plotted in Figure 10a show that its values are higher, especially in the tool feed direction, than those obtained for GFRP under identical conditions. For the highest analyzed cutting speeds and feeds per tooth, the value of the F_x_ force is 63% higher in relation to GFRP because it increases from 58 N (for GFRP) to 95 N (for CFRP). The F_y_ force values also increased, especially in the middle range of the applied cutting speeds and for the highest feeds per tooth (Figure 10b).

The surfaces were plotted based on the formulas for cutting forces in CFRP machining with uncoated carbide tools:F_x_ = 5.59 + 0.39v_c_ − 0.00085v_c_^2^ − 59.77f_z_ + 97.85f_z_^2^ + 0.30v_c_f_z_,(8)F_y_ = −6.96 + 0.48v_c_ − 0.0011v_c_^2^ + 241.45f_z_ − 36.19f_z_^2^ − 0.041v_c_f_z_, (9)

An analysis of the cutting force components during milling of polymer composites with carbon fibers using TiAlN-coated carbide tools shows that, just like for GFRP, the most significant F_x_ cutting force components occur at the highest cutting speed and feed per tooth (Figure 12a). The maximum values of F_x_ reach 133 N. This is a comparable value to that obtained during GFRP machining. This means that coated carbide tools are more versatile for machining various types of polymer composites. An analysis of the F_y_ cutting force component in CFRP machining reveals that its value increases in the middle range of cutting speeds (Figure 12b). This is the opposite trend to that observed during GFRP machining, where the middle range of cutting speeds caused a decrease in the tangential force F_y_. Carbon fiber composites are more durable and cause greater resistance during machining. Glass fiber composites have better machinability, resulting in lower forces. The conclusions regarding average cutting speed ranges are based on differences in the types of reinforcing fibers used

The surfaces in Figure 12a,b were plotted based on Formulas (10) and (11) for the cutting forces during CFRP machining with TiAlN-coated carbide tools:F_x_ = 23.74 + 0.16v_c_ − 0.00026v_c_^2^ + 85.72f_z_ − 137.65f_z_^2^ + 0.37v_c_f_z_,(10)F_y_ = −11.72 + 0.25v_c_ − 0.00052v_c_^2^ + 431.55f_z_ − 258.10f_z_^2^ − 0.076v_c_f_z_,(11)

The final stage of the study involved examining the effect of cutting speed and feed per tooth on the cutting-force components during CFRP milling with polycrystalline diamond inserts (Figure 13a,b). Response surfaces generated from Equation (12) confirm that the feed-force component F_x_ is the highest when the highest values of cutting speed and feed per tooth are applied (Figure 13a). Surfaces based on Equation (13) reveal that, for CFRP, the tangential component F_y_ peaks at the largest feed per tooth combined with intermediate cutting speeds (Figure 13b).

A comparison of the PCD-tool data for GFRP and CFRP shows that the feed-force component F_x_ in CFRP is approximately twice as high as that recorded for GFRP under identical conditions.

Formulas describing the cutting forces in CFRP machining with polycrystalline diamond insert tools:F_x_ = 9.91 + 0.19v_c_ − 0.00031v_c_^2^ − 43.35f_z_ + 108.68f_z_^2^ + 0.31v_c_f_z_, (12)F_y_ = −39.83 + 0.50v_c_ − 0.0011v_c_^2^ + 574.40f_z_ − 509.90f_z_^2^ − 0.02v_c_f_z_, (13)

The analysis of polymeric composite machining using different tools also enables the calculation of the coefficient of determination (R^2^), which determines the quality of model fit to the data. The R^2^ values are listed in Table 3.

An analysis of the data demonstrates that the tools with polycrystalline diamond inserts for both types of materials show the best quality of model fit to the data. The best quality of fit is shown by the tool with polycrystalline diamond inserts in machining glass fiber-reinforced plastics. 

The obtained cutting force results allowed for the determination of cutting resistance. Based on the known values of the cutting force components F_x_ and F_y_ and the cross-section of the removed layer, it was possible to calculate the cutting resistance kc in the x- and *y*-axis directions, which is described by Formula (14): (14)$$k_{c}=\frac{F_{x,y}}{a_{p}\times a_{e}}$$
where F_x,y_ denote the feed and main (tangential) forces, a_p_ is the depth of cut, and a_e_ is the width of the removed layer equal to the cutter’s diameter. The cutting resistances shown in Figure 14 were determined for the highest values of the feed and main (tangential) forces. The *x*-axis indicates the type of tool, where nPVD denotes the uncoated carbide tool, PVD denotes the PVD-coated carbide tool, and PCD denotes the polycrystalline diamond tool.

The results show that the highest cutting resistance was obtained when machining composites with carbon fibers. The only deviation from this trend was observed when the machining process was conducted with coated carbide tools at the main force. A slight increase in the cutting resistance was noticeable with values of 15.2 N/mm^2^ for GFRP and 14.3 N/mm^2^ for CFRP. In the remaining cases, both for the feed force and the main force, the cutting resistance was higher in CFRP machining. The higher strength of CFRP composites can explain this situation.

## 4. Discussion

The problem of machining polymer–matrix composites remains challenging, as shown by this work and previous studies alike. These composites are non-homogeneous materials, which leads to ambiguities in the interpretation of analyses. A composite material consisting of several elements makes it a challenge to clearly determine the effect of tools and their geometry on the components of cutting forces. Previous studies show that the results are often contradictory. Most analyses of the influence of technological parameters on the milling of polymer composites have shown that feed rate has the greatest impact on cutting forces and surface roughness [15]. Researchers have observed that in the case of machining carbon fiber-reinforced plastics, the cutting force increased nonlinearly with increasing feed rate [29]. In addition to the confirmed low feed rates, a fourfold increase in the number of milling cutter flutes has a positive effect on damage and cutting forces [5]. A significant and unambiguous influence of feed on cutting forces was also demonstrated in a study on the machining of carbon fiber composites. However, it was found that cutting forces increase with increasing feed, but also with the helix angle and a large number of tool flutes [15]. The value of the cutting force also depends on the cutting speed, which, when increased, causes an unfavorable increase in tool wear [47]. Studies show that both the feed per blade and the cutting speed in relation to tool wear cause an increase in the cutting force [14]. Another study also showed that damage in the form of delamination, an increase in machining temperature and cutting forces increase with increasing cutting speed [48]. This study investigated the milling of polymer composites with glass and carbon fibers using tools made of different materials and with different geometries. The results enabled the plotting of response surfaces through a two-factor analysis with variable feed per tooth and cutting speed. In effect, it was also possible to establish formulas describing the obtained forces depending on two variable technological parameters of the milling process. In addition, cutting resistances were determined. Previous studies related to polymer composites predominantly involved single-factor analyses. Thanks to the analyses carried out in this article, it was possible to demonstrate a significant influence of feed on cutting force values. Compared to other studies analyzing feed, the research described in this article compares two different composite materials and three tools, which allows for the demonstration of differences in force values. The study also determined the influence of cutting speed on force. The results of the study allowed us to compare the values obtained for all the analyzed variables simultaneously and to show that for both materials, this is a secondary parameter in relation to cutting force. Cutting depth was not analyzed. Previous studies on the machining of polymer composites show that the cutting depth has little or no effect on the quality of the machining. It can be expected that increasing the cutting depth will increase the cutting force. A two-factor analysis was used in this work, so two of the three most important parameters during machining were selected. Polymer composites have a non-homogeneous structure, so it is reasonable that this type of material be investigated taking into account several variable factors. Previous works have also focused on the influence of technological parameters on surface roughness and cutting forces for one type of tool. Among the types of cutting tools analyzed for the machining of polymer composites were tools with polycrystalline diamond (PCD) inserts, tools with a chemically or physically applied thin coating, and tools made of uncoated cemented carbides. This work mainly focused on one type of tool and its effect on machining [49]. This study attempted to analyze the cutting force components for different types of tools, achieving similar values of the cutting force components across various rake angles. The cutting forces are most influenced by the reinforcement in the form of fibers, which are responsible for load transfer. Different values of F_x_ and F_y_ for different tools result from the cutting resistance of the reinforcing fibers. The cutting tool blade cuts through the resin and fibers, but it is the fibers that cause greater cutting resistance. The alternating arrangement of fibers in the fabric forming the prepregs and the distance between the fibers cause the cutting blade to be in constant contact with the reinforcing phase of the composite. There is also a periodic impact of the blade on successive adjacent fibers. Machining with uncoated and coated carbide tools causes wear in the form of abrasion of the rake face. Using tools with polycrystalline diamond plates causes chipping on the corners. Chipping of diamond tools is caused by the fibers in the composite.

In summary, the measurable effects of the presented research are to show the influence of parameters and tools on cutting forces shown on response surfaces in a two-factor analysis. The research also made it possible to determine equations and cutting resistances for all analyzed cutting forces at constant input factors. 

## 5. Conclusions

The paper presented the results of machining tests on polymer–matrix composites using three milling tools made of different materials and having different rake angles. A response surface methodology (RSM) combined with analysis of variance (ANOVA) was applied, with feed per tooth and cutting speed as the independent variables. Milling experiments were conducted seven times for each of the tested surfaces. The study aimed to determine the response surface for tools used in the processing of polymer composites and to define mathematical models. The models derived from the research enable the determination of the feed and the main (tangential) force for variable technological parameters of milling for the cutting tools used in composite machining.

Important conclusions and novelties of this study are as follows:

It is possible to obtain similar values of the main force F_y_ using tools made of uncoated cemented carbides, coated cemented carbides and polycrystalline diamond. For both types of composite materials and different rake angles, the maximum main force ranged from 140 N to 180 N. The results of the F_x_ feed force tests showed that the range of maximum values was from 46 N to 133 N. By taking into account only the cutting force components, it is possible to use cheaper tools with large rake angles as an alternative to expensive polycrystalline diamond tools. This is important for the sustainable development concept in terms of cost reduction and generating savings in the production process.Results of the polymer composite cutting tests showed that of the two analyzed parameters, the feed per tooth had the most significant impact on the feed and main (tangential) forces. The test results show that low feed rates are beneficial for achieving low cutting forces for both composite materials tested. The influence of cutting speed on the forces is less significant, but in most cases, it is recommended to use low and high cutting speeds.An analysis of the determination coefficients showed that all values exceeded 0.90 for the main force; however, for the polycrystalline diamond tool, they were the highest for both materials (accuracy of the specified models ranged 0.98–0.99).Cutting resistance calculations showed that the carbon fiber-reinforced composite was more difficult to machine. The only exception occurred when a tool made of coated cemented carbides was used, where the value of the main force was higher for the composite with carbon fibers. For the applied machining parameters, the cutting resistance ranged from 3.8 N/mm^2^ to 15.5 N/mm^2^.

Overall, the results have shown that polymeric composites are difficult-to-cut materials. In addition, the tool tests pose a need for thorough examination of different rake angles for one type of tool, to find an optimal rake angle in terms of reduced manufacturing costs and rational use of tool resources. Additionally, it is essential to measure surface quality obtained with different tools. A constant value of corner rounding was used in the tests, and the composite material had a fibrous structure. Future works should also attempt to determine the effect of tool rounding on the cutting forces and surface quality. Moreover, it is also crucial that the obtained results be validated using the response surface determination method, yet based on a central, compositional, wall-centered plan.

## Figures and Tables

**Figure 1 materials-18-03017-f001:**
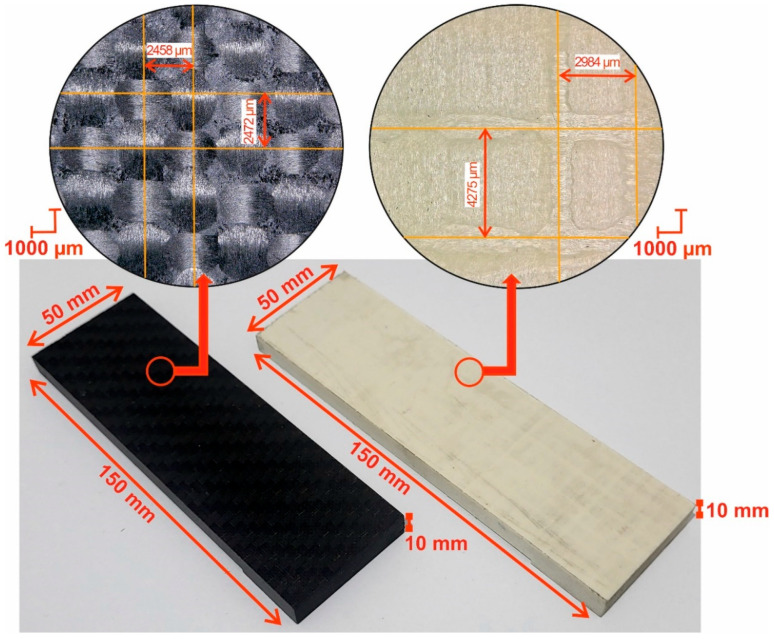
Samples with a cross-linked polymer matrix.

**Figure 2 materials-18-03017-f002:**
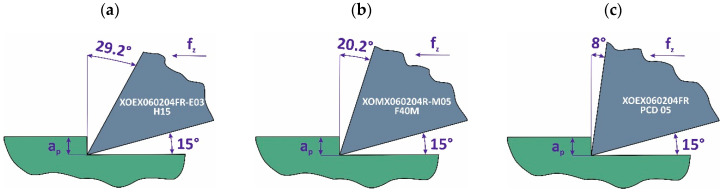
Rake and clearance angles of the tools with different inserts: uncoated cemented carbide (**a**), TiAlN-coated cemented carbide (**b**), polycrystalline diamond (**c**).

**Figure 3 materials-18-03017-f003:**
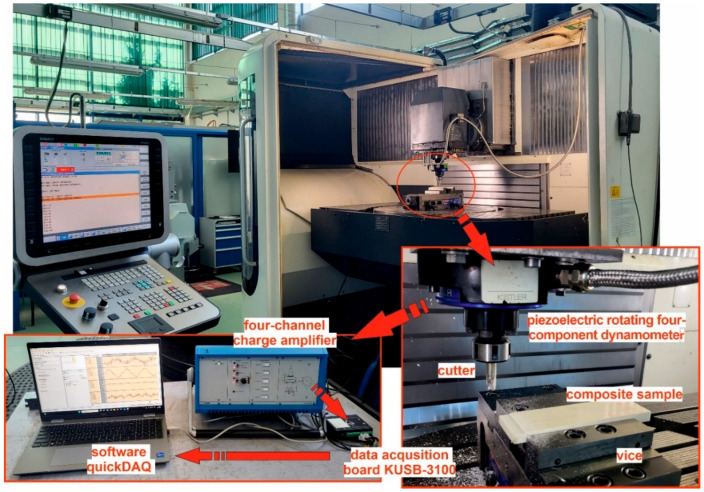
Experimental setup.

**Figure 4 materials-18-03017-f004:**
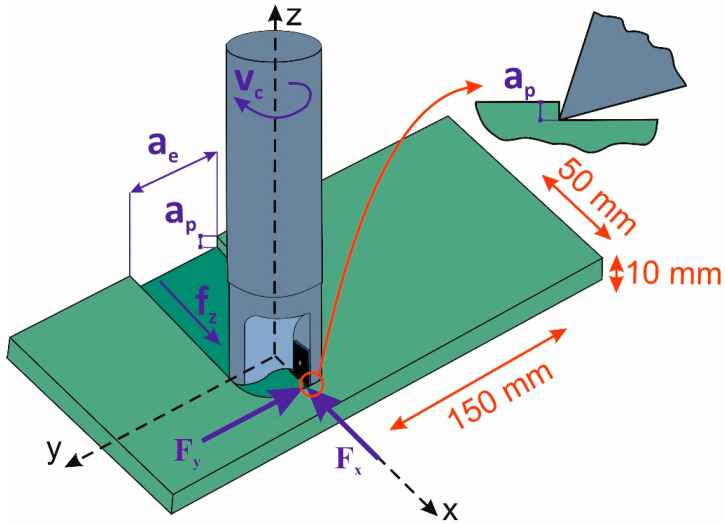
Scheme of cutting kinematics and force distribution.

**Figure 5 materials-18-03017-f005:**
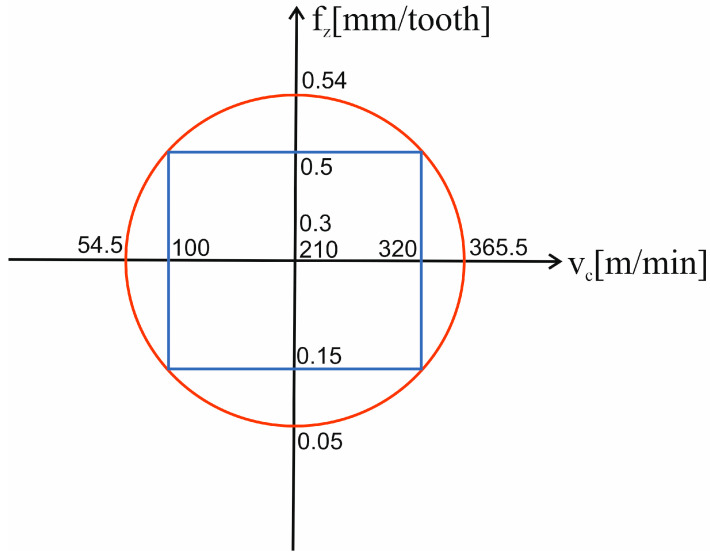
Schematic view of a rotatable design.

**Figure 6 materials-18-03017-f006:**
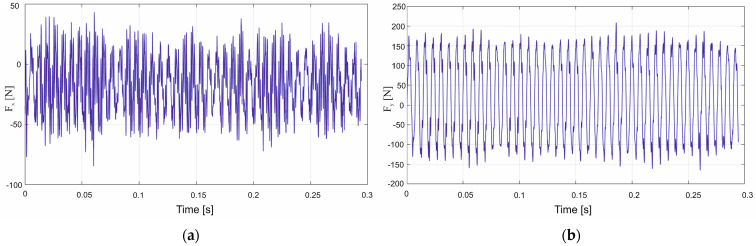
Curves of the cutting force components F_x_ (**a**) and F_y_ (**b**) during GFRP machining by polycrystalline diamond tools.

**Figure 7 materials-18-03017-f007:**
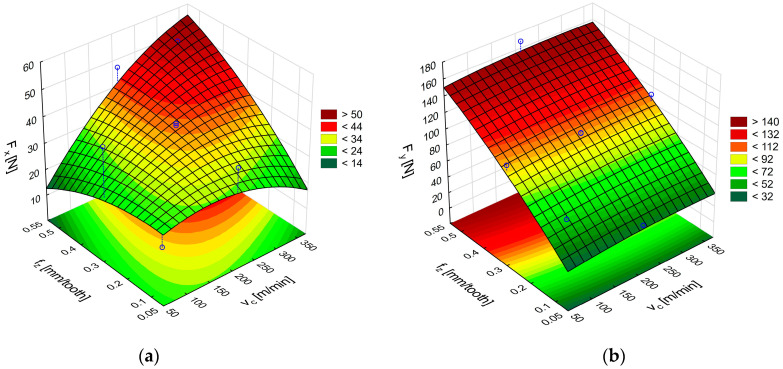
Influence of cutting speed and feed per tooth on the cutting force components F_x_ (**a**) and F_y_ (**b**) during GFRP machining with uncoated carbide tools.

**Figure 8 materials-18-03017-f008:**
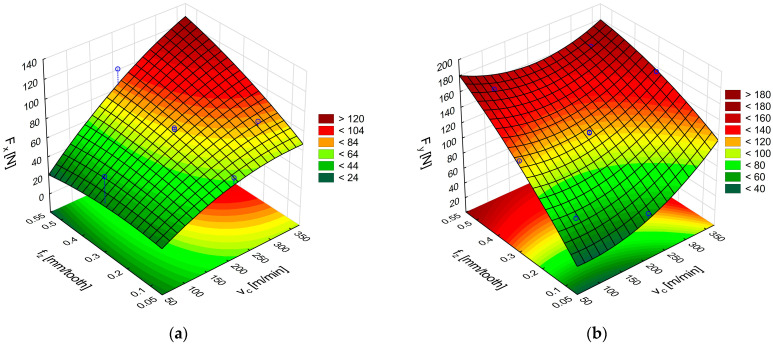
Influence of cutting speed and feed per tooth on the cutting force components F_x_ (**a**) and F_y_ (**b**) during GFRP machining with TiAlN-coated carbide tools.

**Figure 9 materials-18-03017-f009:**
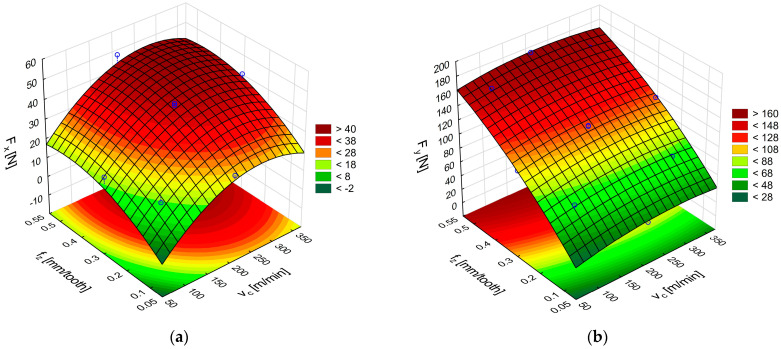
Influence of cutting speed and feed per tooth on the cutting force components F_x_ (**a**) and F_y_ (**b**) during GFRP machining using tools with polycrystalline diamond inserts.

**Figure 10 materials-18-03017-f010:**
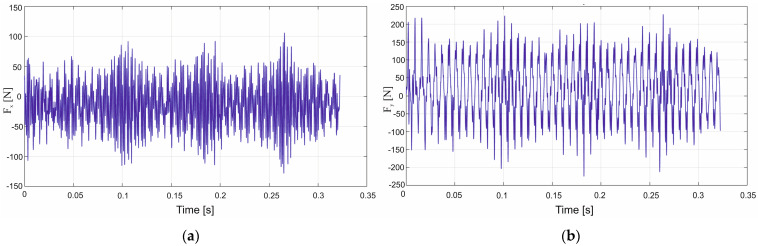
Curves of the cutting force components F_x_ (**a**) and F_y_ (**b**) during CFRP machining with polycrystalline diamond tools.

**Figure 11 materials-18-03017-f011:**
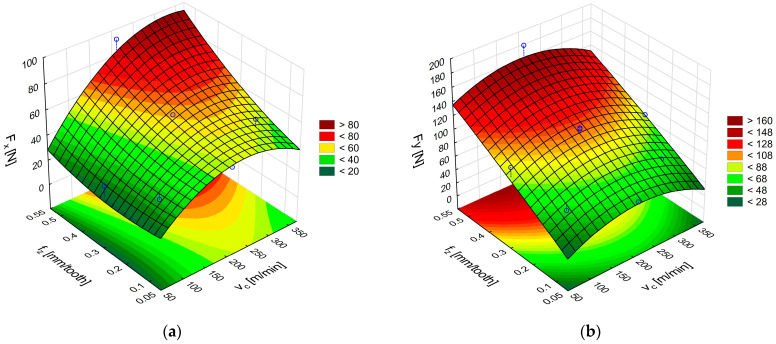
Influence of cutting speed and feed per tooth on the cutting force components F_x_ (**a**) and F_y_ (**b**) during CFRP machining with uncoated carbide tools.

**Figure 12 materials-18-03017-f012:**
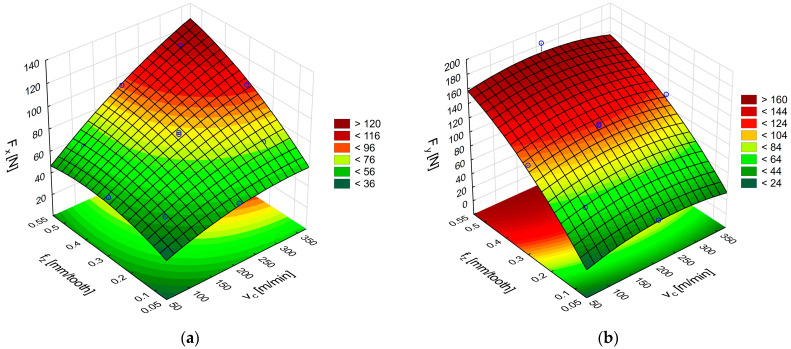
Influence of cutting speed and feed per tooth on the cutting force components F_x_ (**a**) and F_y_ (**b**) during CFRP machining with TiAlN-coated carbide tools.

**Figure 13 materials-18-03017-f013:**
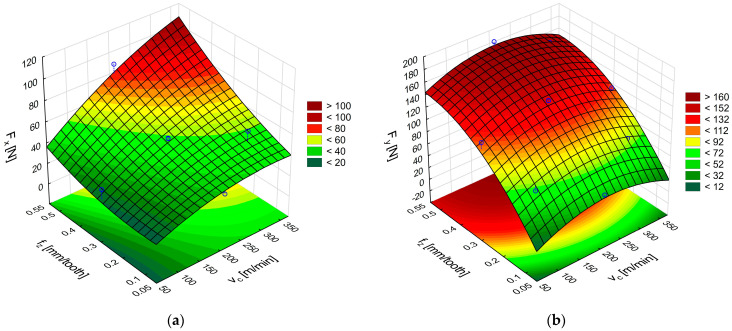
Influence of cutting speed and feed per tooth on the cutting force components F_x_ (**a**) and F_y_ (**b**) during CFRP machining using tools with polycrystalline diamond inserts.

**Figure 14 materials-18-03017-f014:**
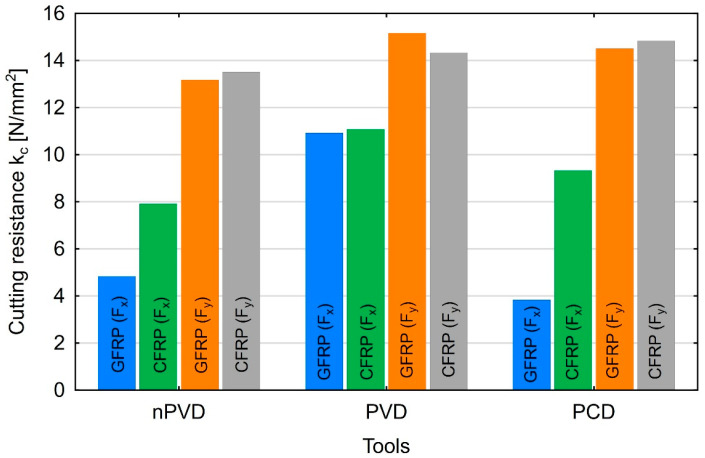
Cutting resistance in the *x* and *y* directions for GFRP and CFRP when using three types of tools.

**Table 1 materials-18-03017-t001:** Tool specifications.

	Tool		
Parameter	XOEX060204FR-E03	XOMX060204R-M05	XOEX060204FR
Insert rake angle	29.2°	20.2°	8°
Grade	H15	F40M	PCD 05
Clearance angle major	15°	15°	15°
Cutting edge effective length	6.0 mm	5.5 mm	2.5 mm
Insert thickness	2.45 mm	2.45 mm	2.45 mm
Insert width	4.1 mm	4.1 mm	4.1 mm
Corner radius	0.4 mm	0.4 mm	0.4 mm

**Table 2 materials-18-03017-t002:** Technological parameters of the milling process.

Configuration	Cutting Speed v_c_ [m/min]	Feed per Tooth f_z_ [mm/tooth]	Depth of Cut a_p_ [mm]
1	100	0.15	1
2	100	0.5	1
3	320	0.15	1
4	320	0.5	1
5	54.5	0.3	1
6	365.5	0.3	1
7	210	0.05	1
8	210	0.54	1
9	210	0.3	1
10	210	0.3	1

**Table 3 materials-18-03017-t003:** Coefficient of determination for the tested tools.

Material of Tool (Composite)	F_x_	F_y_
uncoated carbide (material: GFRP)	0.39	0.96
coated carbide (material: GFRP)	0.74	0.98
polycrystalline diamond (material: GFRP)	0.93	0.99
uncoated carbide (material: CFRP)	0.91	0.92
coated carbide (material: CFRP)	0.99	0.97
polycrystalline diamond (material: CFRP)	0.90	0.98

## Data Availability

The raw data supporting the conclusions of this article will be made available by the authors on request.

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
