# Peer review of "Modeling and Cutting Mechanics in the Milling of Polymer Matrix Composites"

_materials, 2025, doi:10.3390/ma18133017_

Round 1

Reviewer 1 Report

Comments and Suggestions for Authors

The paper is devoted to the study of the mechanics of cutting polymer composite materials during milling and modeling of cutting force components using the response surface method (RSM). The relevance of the study is due to the widespread use of composite materials in aerospace, automotive and other high-tech industries, where the requirements for accuracy and quality of processing are particularly high. Due to the heterogeneous structure of these materials, the choice of a cutting tool and processing parameters is quite complex. The importance of this study lies in the experimental comparison of three types of tools with different geometry and material of the cutting part, as well as in the construction of mathematical models that allow predicting cutting forces. The obtained dependencies demonstrate the possibility of achieving comparable cutting force indicators using less expensive tools, which is of practical value for increasing the economic efficiency of production. The work contributes to the optimization of mechanical processing of difficult-to-machine composite materials.
However, the article needs revision.
1. Lines 66–67 state that increasing the cutting speed worsens the surface quality. However, later (pp. 98–99) it is stated that high speed and low feed contribute to low roughness. It is necessary to clarify the context in which this is true in order to avoid an internal contradiction.
2. Lines 87–92 note that milling defects can lead to complete disqualification of the material. It would be useful to indicate which defects are most critical and whether they were recorded within the framework of this study.
3. Lines 111–121 discuss the factors influencing cutting forces. It would be necessary to more clearly separate the influence of the physical characteristics of the material and the tool geometry, so that these dependencies can be more easily followed in the analysis later.
4. Lines 169–175 formulate the objective of the study - to determine the parameters most favorable in terms of minimizing cutting forces. However, the conclusions (pp. 478–487) focus mainly on comparing different tools. Either the formulation of the objective or the conclusions should be adjusted to achieve complete consistency.
5. In equation (1) on page 301, the feed coefficient has an abnormally high value compared to other variables. It should be explained whether this is due to high sensitivity or to the peculiarities of data normalization.
6. In lines 314–316, a conclusion is made about the importance of the inclination angle, but this is not supported by an analysis of deviations or gradients. It would be useful to supplement the visualization with gradients of forces for different geometries.
7. In lines 343–353, the authors make an important conclusion about the comparability of forces for cheap and expensive tools. However, no economic evaluation is given. Even a brief comparison of the cost of the tool per unit of cutting would significantly increase the practical significance of the work.
8. Equation (7) on page 376 and equation (9) on page 395 have opposite signs for the feed parameter. An explanation is needed whether this is due to differences in tool geometry, material, or an error. 9. Lines 381–386 note the difference in the behavior of Fy between GFRP and CFRP. It would be useful to supplement the analysis with microstructural differences, since the results themselves require an explanation at the material level.
10. Lines 449–467 demonstrate successful implementation of the multivariate approach. However, only the depth of cut is mentioned as a fixed variable. It would be useful to explain why it was not varied and what effect it may have.
11. Lines 478–486 make an important point about the possibility of using low-cost tools. However, no results are given for wear or number of passes. This reduces the practical validity of the recommendation.

Author Response

Responses to comments from Reviewer

We would like to thank the Reviewer for the time spent on carefully reviewing this work and for their valuable deep insight and comments. We feel that this paper is now clearer, more thoroughly discussed and better-referenced. The work has been revised to address the reviewer suggestion. Please find hereafter a point-by-point reply to the comments and suggestions. Red words in article indicate changes (“track changes” option) from the original text of the manuscript. 

General comment from Reviewer:  The paper is devoted to the study of the mechanics of cutting polymer composite materials during milling and modeling of cutting force components using the response surface method (RSM). The relevance of the study is due to the widespread use of composite materials in aerospace, automotive and other high-tech industries, where the requirements for accuracy and quality of processing are particularly high. Due to the heterogeneous structure of these materials, the choice of a cutting tool and processing parameters is quite complex. The importance of this study lies in the experimental comparison of three types of tools with different geometry and material of the cutting part, as well as in the construction of mathematical models that allow predicting cutting forces. The obtained dependencies demonstrate the possibility of achieving comparable cutting force indicators using less expensive tools, which is of practical value for increasing the economic efficiency of production. The work contributes to the optimization of mechanical processing of difficult-to-machine composite materials.

Response: We would like to thank the Reviewer for his opinion. We feel to be obligated to answer for points mention in the review. The paper has been modified and improved. We believe that now is clearer.

Comment 1:  Lines 66–67 state that increasing the cutting speed worsens the surface quality. However, later (pp. 98–99) it is stated that high speed and low feed contribute to low roughness. It is necessary to clarify the context in which this is true in order to avoid an internal contradiction.

Response 1: Thank you for your suggestion regarding the explanation of this phenomenon. In the case of polymer composite machining, most studies show that cutting speed has a positive effect on surface quality by reducing roughness parameters. However, there are some studies that show that cutting speed can cause a deterioration in surface quality. Some conflicting studies are the result of the difficult machinability of polymer composites, which are heterogeneous materials. However, in order to avoid any doubt and internal contradiction regarding the overall influence of cutting speed on surface quality, the statement on roughness has been removed.

Comment 2: Lines 87–92 note that milling defects can lead to complete disqualification of the material. It would be useful to indicate which defects are most critical and whether they were recorded within the framework of this study.

Response 2: Thank you for this suggestion. The authors studied the delamination of polymer composites. The aforementioned study examined the delamination of the upper layers of the composite material, which leads to the rejection of components. According to the authors, the results of the study allow for the adjustment of the processing methodology in order to reduce the number of such defects. Additional information has been added to the article.

Comment 3: Lines 111–121 discuss the factors influencing cutting forces. It would be necessary to more clearly separate the influence of the physical characteristics of the material and the tool geometry, so that these dependencies can be more easily followed in the analysis later.

Response 3: The paper supplements information on the influence of factors on cutting forces. It explains how cutting parameters affect cutting forces. In addition, it explains that the properties of the workpiece material influence cutting forces. In the case of composites, the type of fiber used is an important factor influencing cutting forces. Carbon fibers, compared to glass fibers, offer greater cutting resistance, resulting in higher forces. In terms of geometry, the corner rounding radius is important. Large corner rounding radii result in higher cutting forces due to the larger cross-sectional area of the cut layer. This information is supplemented in the article.

Comment 4: Lines 169–175 formulate the objective of the study - to determine the parameters most favorable in terms of minimizing cutting forces. However, the conclusions (pp. 478–487) focus mainly on comparing different tools. Either the formulation of the objective or the conclusions should be adjusted to achieve complete consistency.

Response 4: Thank you for your valid observation. Additional information on the influence of parameters on cutting forces has been added to the conclusions. It has been added that "low feed rates are beneficial for achieving low cutting forces for both composite materials tested. The influence of cutting speed on the forces is less significant, but in most cases it is recommended to use low and high cutting speeds."

Comment 5: In equation (1) on page 301, the feed coefficient has an abnormally high value compared to other variables. It should be explained whether this is due to high sensitivity or to the peculiarities of data normalization.

Response 5: The equations in the article are determined on the basis of response surfaces plotted based on the cutting parameters used. The feed coefficient in equation (1) is related to the sharpest edge of the blade (29.2°) for an uncoated tool and the shear angle for a glass fiber reinforced plastics. It is also related to the conditions of interaction between the material and the geometry of the blade with the workpiece. The selection of a rational shear angle in terms of cutting mechanics requires further exploratory research. Further studies are planned using tools with different rounding radii and different geometries (rake and clearance angles).

Comment 6: In lines 314–316, a conclusion is made about the importance of the inclination angle, but this is not supported by an analysis of deviations or gradients. It would be useful to supplement the visualization with gradients of forces for different geometries.

Response 6: Thank you for noticing this conclusion. The paper analyzed the influence of cutting parameters and rake angle on cutting forces. In order not to mislead readers, the statement “This leads to a conclusion that with these two types of tools, the rake angle has a greater influence on the cutting force components than the type of tool material” has been removed.

Comment 7: In lines 343–353, the authors make an important conclusion about the comparability of forces for cheap and expensive tools. However, no economic evaluation is given. Even a brief comparison of the cost of the tool per unit of cutting would significantly increase the practical significance of the work.

Response 7: This is a very interesting issue. The cost of a single cutting insert made of polycrystalline diamond is approximately €100, while the cost of a carbide insert is approximately €12. Of course, I agree with the reviewer's opinion that it would be useful to provide the cost of the tool per unit of cutting, but unfortunately I do not have such data. However, this is an interesting direction for further research, in which the wear of different cutting inserts would be compared under the same machining conditions.

Comment 8: Equation (7) on page 376 and equation (9) on page 395 have opposite signs for the feed parameter. An explanation is needed whether this is due to differences in tool geometry, material, or an error.

Response 8: Thank you for reading the article carefully. The equations are the result of the response planes plotted. I have checked these equations again and they are correct. The differences in signs are due to the specific nature of all the input factors of the study. This is mainly related to factors such as the interaction between the geometry, tool material, and coating of the cutting edge with the workpiece material. The range of available tool grades and types is limited, so further research is planned using custom-made tools that are not part of the standard tool range. Obtaining special custom-made tools will allow for a better understanding of the cutting mechanics of polymer composites.

Comment 9: Lines 381–386 note the difference in the behavior of Fy between GFRP and CFRP. It would be useful to supplement the analysis with microstructural differences, since the results themselves require an explanation at the material level.

Response 9: Thank you for your helpful suggestion regarding the materials. The information related to the processed materials has been supplemented. The article now includes the following statement: "Carbon fiber composites are more durable and cause greater resistance during machining. Glass fiber composites have better machinability, resulting in lower forces. The conclusions regarding average cutting speed ranges are based on differences in the types of reinforcing fibers used.

Comment 10: Lines 449–467 demonstrate successful implementation of the multivariate approach. However, only the depth of cut is mentioned as a fixed variable. It would be useful to explain why it was not varied and what effect it may have.

Response 10: Thank you for this observation. The following information has been added to the “Discussions” section of the paper: "Previous studies on the machining of polymer composites show that the cutting depth has little or no effect on the quality of the machining. It can be expected that increasing the cutting depth will increase the cutting force. A two-factor analysis was used in this work, so two of the three most important parameters during machining were selected."

Comment 11: Lines 478–486 make an important point about the possibility of using low-cost tools. However, no results are given for wear or number of passes. This reduces the practical validity of the recommendation.

Response 11: Thank you for this suggestion. Tool wear was not analyzed in the study. The tests were conducted seven times, so the number of passes on each surface was seven. Information on the number of attempts has been added to the conclusions.

We appreciate for Reviewer warm work earnestly, and hope that the corrections will meet with approval. Once again, we thank you very much for your comment and suggestion.

Yours sincerely,

Krzysztof CiecielÄ…g

Andrzej Kawalec

Michał Gdula

Piotr Żurek

Reviewer 2 Report

Comments and Suggestions for Authors

The study investigates the problem of modeling cutting‑force components by  means of response surface methodology and reports the results of an investigation into the impact of machining parameters on the cutting mechanics of polymer‑matrix composites. The novelty of this study is the modeling of cutting forces and the determination of  mathematical models of these forces. The models describe the values of forces depending on the milling parameters. In addition, the cutting resistance of the composites was determined. The influence of the material and rake angle of individual tools on the cutting force  components was also determined. Some contents need to be revised.

  1. In the abstract, the description of results is very little, only three lines. And the author should conclude more results here.
  2. In line 225-237, the grammar sentences need to be improved.
  3. How to obtain the cutting force components Fx and Fy curves? The solving method of experiment data  should be given in detail.
  4. The glass fiber function mechanism in glass fiber‑reinforced plastics need to be explained in detail.
  5. The conlusion should be simplified further, and the graphic abtract should be given.

Comments on the Quality of English Language

The study investigates the problem of modeling cutting‑force components by  means of response surface methodology and reports the results of an investigation into the impact of machining parameters on the cutting mechanics of polymer‑matrix composites. The novelty of this study is the modeling of cutting forces and the determination of  mathematical models of these forces. The models describe the values of forces depending on the milling parameters. In addition, the cutting resistance of the composites was determined. The influence of the material and rake angle of individual tools on the cutting force  components was also determined. Some contents need to be revised.

  1. In the abstract, the description of results is very little, only three lines. And the author should conclude more results here.
  2. In line 225-237, the grammar sentences need to be improved.
  3. How to obtain the cutting force components Fx and Fy curves? The solving method of experiment data  should be given in detail.
  4. The glass fiber function mechanism in glass fiber‑reinforced plastics need to be explained in detail.
  5. The conlusion should be simplified further, and the graphic abtract should be given.

Author Response

Responses to comments from Reviewer

We would like to thank the Reviewer for the time spent on carefully reviewing this work and for their valuable deep insight and comments. We feel that this paper is now clearer, more thoroughly discussed and better-referenced. The work has been revised to address the reviewer suggestion. Please find hereafter a point-by-point reply to the comments and suggestions. Red words in article indicate changes (“track changes” option) from the original text of the manuscript. 

General comment from Reviewer: The study investigates the problem of modeling cutting‑force components by  means of response surface methodology and reports the results of an investigation into the impact of machining parameters on the cutting mechanics of polymer‑matrix composites. The novelty of this study is the modeling of cutting forces and the determination of  mathematical models of these forces. The models describe the values of forces depending on the milling parameters. In addition, the cutting resistance of the composites was determined. The influence of the material and rake angle of individual tools on the cutting force  components was also determined. Some contents need to be revised.

Response: We would like to thank the Reviewer for his opinion. We feel to be obligated to answer for points mention in the review. The paper has been modified and improved. We believe that now is clearer.

Comment 1: In the abstract, the description of results is very little, only three lines. And the author should conclude more results here.

Response 1: Thank you for your suggestion. The abstract provides more detailed research results. Information on the obtained ranges of the tangential cutting force component Fy has been added. Information on the obtained results of cutting resistance has also been added.

Comment 2: In line 225-237, the grammar sentences need to be improved.

Response 2: Thank you for this suggestion. The article has been reviewed by a native English speaker. Changes have also been made to the suggested paragraphs.

Comment 3: How to obtain the cutting force components Fx and Fy curves? The solving method of experiment data  should be given in detail.

Response 3: Thank you for this comment. For a detailed discussion of the calculation of experimental data, explanations on how the cutting force component curves were created are provided at the end of subsection “2.3. Method and technological parameters of the milling process.” We added: “The statistical method of determining the response surface (RSM) was used to model the relationships between the cutting process parameters and the cutting force components. RSM-based research plans refer to the determination of a response surface based on a general equation of the type:

Y=β01X12X211(X1)222(X2)212X1X2

As a result, a model is fitted to the experimental output values that captures the influence of the main input values (X1, X2), the interactions between the input values (X1X2), and the quadratic terms (X12, X22). β0 is the intercept of the arithmetic averages of all the quantitative outcomes. Parameters β1, β2, β11, β22 and β12 are the coefficients computed from the experimental values of Y. Regression statistics commonly used in mechanical engineering, such as e.g., ANOVA, was used to identify the most influential factors, create suitable mathematical model and evaluate its fit to the measurement data.”

Comment 4: The glass fiber function mechanism in glass fiber‑reinforced plastics need to be explained in detail.

Response 4: Thank you. In the article has been added an information about the function of fibers and the importance of fabrics in machining. Information: "Glass and carbon fibers reinforce polymer composites. They are arranged in a 0°-90° pattern to form a fabric which, when saturated with epoxy resin, forms a prepreg. This arrangement ensures balanced material strength, which is particularly important in machining.“ has been added to the subsection: ”2.1. Materials and sample preparation."

Comment 5: The conclusion should be simplified further, and the graphic abstract should be given.

Response 5: Thank you for this suggestion. The conclusions have been simplified and several statements have been removed. The following has been removed: “The study also showed that tool geometry had a significant effect on the main force.” This conclusion is included in the first point of this chapter. The following has also been removed: “On this basis, it can be concluded that” and “have provided interesting conclusions, which” to simplify the conclusions. In point 2 of the conclusions, at the request of another reviewer, information on the recommended cutting parameters to achieve low cutting forces has been added. To improve the readability of the article, a graphical abstract has also been prepared.

We appreciate for Reviewer warm work earnestly, and hope that the corrections will meet with approval. Once again, we thank you very much for your comment and suggestion.

Yours sincerely,

Krzysztof CiecielÄ…g

Andrzej Kawalec

Michał Gdula

Piotr Żurek

Reviewer 3 Report

Comments and Suggestions for Authors

This paper presents the modeling and cutting mechanics in the milling of polymer matrix composites. Some issues need to be addressed as following:

  1. The response surface methodology is utilized to develop the cutting force model, which is based on the experimental results. What’s its mechanism?
  2. The cutting force in milling of the polymer matrix composites is closely related to the structure of the material, such as the angle of the fibers. However, from the equation of the cutting force proposed in this paper, it is found that it is only the function of the cutting parameters, and not related to the material parameters. Please explain it.
  3. The discussion section is too brief, and needs to be improved. The experimental observation needs to be further discussed.
  4. Three different kinds of tools are discussed in this paper. For a certain kind of the composites, it has the most suitable tool. So, what is the significance of this discussion (Figure 14)?
  5. the conclusion of this paper is qualitative, even the few quantitative conclusions proposed in this paper are not universally applicable.

Author Response

Responses to comments from Reviewer

We would like to thank the Reviewer for the time spent on carefully reviewing this work and for their valuable deep insight and comments. We feel that this paper is now clearer, more thoroughly discussed and better-referenced. The work has been revised to address the reviewer suggestion. Please find hereafter a point-by-point reply to the comments and suggestions. Red words in article indicate changes (“track changes” option) from the original text of the manuscript. 

General comment from Reviewer: This paper presents the modeling and cutting mechanics in the milling of polymer matrix composites. Some issues need to be addressed as following.

Response:

We would like to thank the Reviewer for his opinion. We feel to be obligated to answer for points mention in the review. The paper has been modified and improved. We believe that now is clearer.

Comment 1: The response surface methodology is utilized to develop the cutting force model, which is based on the experimental results. What’s its mechanism?

Response 1: Thank you for this comment. For a detailed discussion of the calculation of experimental data, explanations on how the cutting force component curves were created are provided at the end of subsection “2.3. Method and technological parameters of the milling process.” We added: “The statistical method of determining the response surface (RSM) was used to model the relationships between the cutting process parameters and the cutting force components. RSM-based research plans refer to the determination of a response surface based on a general equation of the type:

Y=β01X12X211(X1)222(X2)212X1X2

As a result, a model is fitted to the experimental output values that captures the influence of the main input values (X1, X2), the interactions between the input values (X1X2), and the quadratic terms (X12, X22). β0 is the intercept of the arithmetic averages of all the quantitative outcomes. Parameters β1, β2, β11, β22 and β12 are the coefficients computed from the experimental values of Y. Regression statistics commonly used in mechanical engineering, such as e.g., ANOVA, was used to identify the most influential factors, create suitable mathematical model and evaluate its fit to the measurement data.”.

Comment 2: The cutting force in milling of the polymer matrix composites is closely related to the structure of the material, such as the angle of the fibers. However, from the equation of the cutting force proposed in this paper, it is found that it is only the function of the cutting parameters, and not related to the material parameters. Please explain it.

Response 2: Thank you for this question. This study investigated the milling of polymer composites formed from fabrics with an alternating fiber arrangement of 0°-90°. For both composites, the prepreg consisted only of fabrics. I agree with the reviewer's opinion that the patterns are related only to the cutting parameters because they were the variable factor for the two-factor analysis. In the tests for both composites, a constant material type was assumed.

Comment 3: The discussion section is too brief, and needs to be improved. The experimental observation needs to be further discussed.

Response 3: Thank you for your kind comment. The “Discussion” section has been greatly expanded in the paper. Reference has been made to previous work related to the topic of composite machining. This section also presents a new feature, namely analysis using a response surface and the determination of equations for calculating the cutting force components for the same conditions for both materials tested.

Comment 4: Three different kinds of tools are discussed in this paper. For a certain kind of the composites, it has the most suitable tool. So, what is the significance of this discussion (Figure 14)?

Response 4: Thank you for your question. In this study, three different tools were analyzed, which are successfully used for machining both carbon and glass fiber composites. Of course, depending on the tools used, a surface with specific parameters is created. This article does not analyze the impact of tools and parameters on the quality of the machined surface. This will be the subject of further research. The aim of this work was to compare and determine the cutting resistance using the same input factors. The cutting forces shown in Figure 14 allow the test results to be compared and show that different tools can be used successfully when only the cutting force is taken into account. It was also shown that greater cutting resistance occurs during the machining of carbon fiber reinforced plastics.

Comment 5: The conclusion of this paper is qualitative, even the few quantitative conclusions proposed in this paper are not universally applicable.

Response 5: Thanks for your comments on the conclusions. The paper presents research on the machining of polymer composites using various tools designed for this type of material. The influence of cutting parameters on cutting force values was analyzed using different tools. The results of the study showed that, considering only cutting forces, both expensive tools with diamond inserts and inexpensive carbide tools can be used successfully. The universal application of the research is the conclusion that different tools can be used interchangeably. This is important for the concept of sustainable development in terms of cost reduction and savings in the production process.

We appreciate for Reviewer warm work earnestly, and hope that the corrections will meet with approval. Once again, we thank you very much for your comment and suggestion.

Yours sincerely,

Krzysztof CiecielÄ…g

Andrzej Kawalec

Michał Gdula

Piotr Żurek

Reviewer 4 Report

Comments and Suggestions for Authors

This manuscript investigates the modeling of cutting forces in the milling of glass and carbon fiber-reinforced polymer matrix composites using response surface methodology (RSM). The study systematically explores the influence of cutting parameters and tool types (uncoated carbide, TiAlN-coated carbide, and polycrystalline diamond) on cutting force components (Fx and Fy), and develops mathematical models with high R2 values. The research is relevant and addresses a timely topic with applications in aerospace and automotive manufacturing. The manuscript is generally well structured, and the experiments are carefully designed and executed. However, before it can be accepted for publication, minor revisions are necessary to improve the scientific clarity, strengthen theoretical insights, and enhance the overall impact.

  1. While the RSM-based modeling is valid, the approach is rather conventional and lacks novelty. The manuscript would benefit from a more detailed discussion of the mechanistic understanding behind cutting force variation, especially considering the heterogeneous nature of polymer matrix composites. Please elaborate on why certain tools (e.g., PCD vs. uncoated carbide) behave differently in Fx vs. Fy directions beyond the rake angle. Are there underlying failure mechanisms (e.g., fiber pull-out, matrix cracking) that support these trends?
  2. Discussion needs deeper comparison with existing literature. Some conclusions (e.g., the dominance of feed per tooth over cutting speed) are consistent with prior work. Please provide a more critical comparison with recent studies to highlight your study’s contribution and novelty.
  3. The manuscript is generally readable but should be polished by a native English speaker or professional editor. Some sentences are overly verbose or redundant.

Author Response

Responses to comments from Reviewer

We would like to thank the Reviewer for the time spent on carefully reviewing this work and for their valuable deep insight and comments. We feel that this paper is now clearer, more thoroughly discussed and better-referenced. The work has been revised to address the reviewer suggestion. Please find hereafter a point-by-point reply to the comments and suggestions. Red words in article indicate changes (“track changes” option) from the original text of the manuscript. 

General comment from Reviewer:  This manuscript investigates the modeling of cutting forces in the milling of glass and carbon fiber-reinforced polymer matrix composites using response surface methodology (RSM). The study systematically explores the influence of cutting parameters and tool types (uncoated carbide, TiAlN-coated carbide, and polycrystalline diamond) on cutting force components (Fx and Fy), and develops mathematical models with high R2 values. The research is relevant and addresses a timely topic with applications in aerospace and automotive manufacturing. The manuscript is generally well structured, and the experiments are carefully designed and executed. However, before it can be accepted for publication, minor revisions are necessary to improve the scientific clarity, strengthen theoretical insights, and enhance the overall impact.

Response: We would like to thank the Reviewer for his opinion. We feel to be obligated to answer for points mention in the review. The paper has been modified and improved. We believe that now is clearer.

Comment 1: While the RSM-based modeling is valid, the approach is rather conventional and lacks novelty. The manuscript would benefit from a more detailed discussion of the mechanistic understanding behind cutting force variation, especially considering the heterogeneous nature of polymer matrix composites. Please elaborate on why certain tools (e.g., PCD vs. uncoated carbide) behave differently in Fx vs. Fy directions beyond the rake angle. Are there underlying failure mechanisms (e.g., fiber pull-out, matrix cracking) that support these trends?

Response 1: Thank you for your suggestion. Polymer composites are difficult-to-machine materials with a heterogeneous structure. Many studies analyzing the machining of these materials emphasize that it is a material whose machining causes a nonlinear relationship. Changes in cutting force are the result of the multi-component structure of polymer composites. The cutting forces are most influenced by the reinforcement in the form of fibers, which are responsible for load transfer. Different values of Fx and Fy for different tools result from the cutting resistance of the reinforcing fibers. The cutting tool blade cuts through the resin and fibers, but it is the fibers that cause greater cutting resistance. The alternating arrangement of fibers in the fabric forming the prepregs and the distance between the fibers cause the cutting blade to be in constant contact with the reinforcing phase of the composite. There is also a periodic impact of the blade on successive adjacent fibers. As demonstrated, a tool with a diamond insert with a small rake angle gives similar results to a tool made of uncoated cemented carbide but with a rake angle three times greater. Further work is planned to study the machining of composites in order to analyze the surface after machining and to identify typical damage associated with a specific tool. Previous in-house studies have shown that fiber pull-out occurs mainly at low cutting speeds. The test results show that with increasing feed rate, the number of surface defects in the form of fiber pull-out gradually increases.

Explanations of the changes in cutting forces in contact with the reinforcing fibers of the composite are presented at the end of the Discussion section.

Comment 2: Discussion needs deeper comparison with existing literature. Some conclusions (e.g., the dominance of feed per tooth over cutting speed) are consistent with prior work. Please provide a more critical comparison with recent studies to highlight your study’s contribution and novelty.

Response 2: Thank you for your suggestion. The “Discussion” section has been greatly expanded in the paper. Reference has been made to previous work related to the topic of composite machining. This section also presents a new feature, namely analysis using a response surface and the determination of equations for calculating the cutting force components for the same conditions for both materials tested.

Comment 3: The manuscript is generally readable but should be polished by a native English speaker or professional editor. Some sentences are overly verbose or redundant..

Response 3: Thank you for this suggestion. The article has been reviewed by a native English speaker.

We appreciate for Reviewer warm work earnestly, and hope that the corrections will meet with approval. Once again, we thank you very much for your comment and suggestion.

Yours sincerely,

Krzysztof CiecielÄ…g

Andrzej Kawalec

Michał Gdula

Piotr Żurek

Round 2

Reviewer 1 Report

Comments and Suggestions for Authors

The authors of the article have generally worked out my comments with high quality and in good faith. However, a number of issues remain unresolved or have not been fully resolved.

Add an estimate of the cost of the tool per meter/pass, even if it is approximate.

Conduct or describe observations of tool wear.

Clarify or structure the influence of geometric parameters and microstructural differences on the behavior of forces.

Author Response

Responses to comments from Reviewer

We would like to thank the Reviewer for the time spent on carefully reviewing this work and for their valuable deep insight and comments. We feel that this paper is now clearer, more thoroughly discussed and better-referenced. The work has been revised to address the reviewer suggestion. Please find hereafter a point-by-point reply to the comments and suggestions. Red words in article indicate changes (“track changes” option) from the original text of the manuscript. 

General comment from Reviewer: The authors of the article have generally worked out my comments with high quality and in good faith. However, a number of issues remain unresolved or have not been fully resolved.

Response: We would like to thank the Reviewer for his opinion. We feel to be obligated to answer for points mention in the review. The paper has been modified and improved. We believe that now is clearer.

Comment 1: Add an estimate of the cost of the tool per meter/pass, even if it is approximate.

Response 1: Thank you for your question The estimated cost of machining with tool with polycrystalline diamond insert is €0.02 per meter. In comparison, for an uncoated carbide tool it is €0.006 per meter. Coated carbide tools are €0.01 per meter. These are the estimated costs of machining polymer composites. The quality indicator is tool wear above 0.5 mm.

Comment 2: Conduct or describe observations of tool wear.

Response 2: Thank you for this suggestion. Machining with uncoated and coated carbide tools causes wear in the form of abrasion of the rake face. Using tools with polycrystalline diamond plates causes chipping on the corner. Chipping of diamond tools is caused by the carbon fibers in the composite. Information has been added to the article in “Discussion” chapter.

Comment 3: Clarify or structure the influence of geometric parameters and microstructural differences on the behavior of forces.

Response 3: In the conclusions in point 2, a summary of the influence of milling technological parameters is given. It is described that low feed rates are beneficial for obtaining low cutting forces for composite materials. In the case of cutting speeds, it is recommended to use low and high cutting speeds to obtain low force values. The polymer composites analyzed in this article consist of glass and carbon fibers. Microstructural differences in force behavior are a subject for further analysis. It is worth examining the effect of fiber density on force behavior.

We appreciate for Reviewer warm work earnestly, and hope that the corrections will meet with approval. Once again, we thank you very much for your comment and suggestion.

Yours sincerely,

Krzysztof CiecielÄ…g

Andrzej Kawalec

Michał Gdula

Piotr Żurek

Reviewer 2 Report

Comments and Suggestions for Authors

The revised manuscript can be published in Journal.

Author Response

Dear Reviewer,

Thank you and we appreciate for Reviewer work.

Yours sincerely,

Krzysztof CiecielÄ…g

Andrzej Kawalec

Michał Gdula

Piotr Żurek

Reviewer 3 Report

Comments and Suggestions for Authors

The proposed comments have been well addressed. No more comments.

Author Response

(The authors gave the same response as above.)
